# Addressing Inequity: Evaluation of an Intervention to Improve Accessibility and Quality of a Green Space

**DOI:** 10.3390/ijerph16245015

**Published:** 2019-12-10

**Authors:** Kirsti S. Anthun, Ruca Elisa Katrin Maass, Siren Hope, Geir Arild Espnes, Ruth Bell, Matluba Khan, Monica Lillefjell

**Affiliations:** 1Department of Neuromedicine and Movement Science, Faculty of Medicine and Health Sciences, Norwegian University of Science and Technology, 7491 Trondheim, Norway; Ruca.e.maass@ntnu.no (R.E.K.M.); siren.hope@ntnu.no (S.H.); Monica.lillefjell@ntnu.no (M.L.); 2Department of Public Health and Nursing, Center for Health Promotion Research, Faculty of Medicine and Health Sciences, Norwegian University of Science and Technology, 7491 Trondheim, Norway; geir.arild.espnes@ntnu.no; 3Institute of Health Equity, Department for Epidemiology & Public Health, University College London, London WC1E 7HB, UK; r.bell@ucl.ac.uk (R.B.); matluba.khan@ucl.ac.uk (M.K.)

**Keywords:** green space, suburban area, equity, use, accessibility, quality, health

## Abstract

Green space areas offer several benefits that support our physical, psychological, and social health. However, the level of engagement with green space areas may not be the same across population groups. Using a mixed-method research design, we investigated the use of a green space area and whether and how the area was beneficial for health, social inclusion, and physical activity for all socioeconomic groups in a suburban area in Norway. The study showed significantly increased use of the area from 2015–2018 and that users belonged to different socioeconomic groups. The motivation for using the area was the opportunity to experience nature and to interact socially. While no significant changes in self-rated health, life satisfaction, or levels of physical activity were found, the study indicates that factors such as location, availability, and designated places for social interaction are important motivating factors for use. Users from the lower socioeconomic groups were among the frequent users but were also the least satisfied with the quality and availability of the path. Our findings call for closer consideration of the location and availability of green spaces and that including places for social interaction and relaxation can contribute to increased use of green spaces.

## 1. Introduction

The associations between green space, health, and well-being are well described and have been outlined in numerous publications [1,2,3,4,5,6,7,8]. Ensuring that public green spaces are available and accessible for all population groups can promote contact with nature, social interaction and physical activity (e.g., walking, running, and cycling), and have a positive impact on a range of health outcomes [3,4,9,10,11,12]. However, some studies have also found that not all public green spaces facilitate physical activity [13,14]; thus, it can be argued that public green spaces have the potential to promote physical activity, but certain characteristics are needed for this to be realised. Public green spaces can, for instance, pose safety threats for some groups [15,16,17] and users’ intended purpose for using green spaces are varied [18,19]; some people go there for relaxation, stress reduction, and to obtain peace [18].

Some evidence suggests that access to public green spaces can be linked to social indicators of health, such as community identity and relationship networks [20,21], which again may support mental renewal and enhance community satisfaction, sense of community, and community attachment among residents [21,22,23,24,25], and that aesthetic surroundings can improve place attachment, or emotional bonds to a location [26].

Research on public green spaces has also devoted much attention towards walking and cycling accessibility in neighbourhoods and the area characteristics around these structures, such as the quality of surface, cleanliness, trees, benches, parks, and safety [27,28,29]. Appropriate facilities for walking and biking can promote physical activity both for transport and leisure as well as increased contact among neighbours [30,31]. Furthermore, living in greener neighbourhoods has been found to create a stronger green space–health association [32,33] as well as lower levels of health inequality related to income deprivation [34]. Studies have also indicated that socioeconomically disadvantaged populations gain particular health benefits from greater contact with and use of green space areas [5,10,11,12,35,36,37,38], most likely because this group generally has poorer health and lives in areas with more environmental problems [35]. This group also tends to be more dependent on local resources [9,11,12,34,39] since they have fewer opportunities to travel away from their neighbourhoods of residence. Thus, equitable access to local resources, such as green spaces, can be a potential way to moderate health inequalities associated with socioeconomic deprivation [5,10,11,12,23,36,40,41,42]. It is also the case that providing public green spaces that are inaccessible for socioeconomically disadvantaged groups might result in widening both health and social inequalities [12,41,42,43].

Although results so far show evidence that supports the use of certain urban green space interventions for health, social, and environmental benefits, we still lack knowledge about whether such benefits are equitable across population groups [44]. The overarching aim of this study was to discover who uses the green space areas and their motivation for doing so, as well as to provide knowledge on whether and how green spaces are beneficial for health, social inclusion, and physical activity for all inhabitants in the community. The study was part of the EU Horizon 2020 INHERIT project (2016–19) working across sectors to achieve triple-wins; improvement in health, equity, and environmental sustainability. The evaluation was grounded in the INHERIT Model [45], which combines determinants of environmental health and health equity with theory on behavioural change.

### Context

This study was conducted in the context of a municipality in Norway with a population of approximately 14,000 displaying a variety of settlements ranging from urban to rural. The Malvik path is a green space area with a three-km-long cycling and walking path along the coast in the municipality of Malvik. Before 2015, the path was only a narrow footpath along an old, abandoned railway track. The footpath connected a residential area with the municipal city centre and went along the coast. It was used by inhabitants moving between the community at one end and the municipal city centre at the other, as well as by those who wanted to go fishing or swimming in the sea. In 2012 it was decided, based on feedback from the inhabitants of the municipality, to build a more accessible walking/biking path on the old railway track, thus enabling improved access to both green space and the sea. During the process of planning, designing, and implementing the path, the municipality administration used inclusive methods to involve and engage the population. The new, improved path was officially opened in June 2016, though by then it had been opened for use for two months. Along the new path, which is much wider than the former footpath, there are now designated places for social interaction, fishing, playing, and barbecuing. Benches have been placed along the path, inviting people to rest and admire the scenery. To create a better sense of place, information boards on local wildlife and history and a selection of historical artefacts have been installed at the side of the path [46,47]. Moreover, the path has been designed according to the principles of universal design (defined by the Norwegian Disability Act of 2005), meaning that the design and composition is so that it can be accessed and used by all people regardless of their age, size, ability, or disability. The surface is flat and firm, and at each end, there are toilet facilities and designated areas for parking.

## 2. Materials and Methods

A mixed-method research design [48], combining a quantitative component consisting of counting data, questionnaire surveys, and registry data with a qualitative component of structured interviews, is applied to provide relevant and sensitive evidence on use, user groups, motivation for use, and possible social and health benefits. This included: (1) a digital counter that registers the number of passings on the path each day; (2) a population survey conducted among inhabitants in the municipality of Malvik; (3) structured, on-site interviews with users of the path; and (4) a short survey completed by users of the path.

### 2.1. Digital Counter

A digital counter, PYRO box from Eco Counter (https://www.eco-compteur.com/en/resources/), that registers each passing at the path was set up in October 2015, prior to the official opening of the renewed path (June 2016). The PYRO-Box (Eco-Counter, Montreal, QC, Canada) is a people counter using passive-infrared, pyroelectric technology that is suitable to count both cyclists and pedestrians. The digital counter has registered the number of passings on the path each day since October 2015, and all registrations from then through to the 31 December 2018 were included in the study (altogether 37.5 months). By use of Eco-Visio data analysis platform 5 (https://www.eco-compteur.com/en/produits/eco-visio-range/eco-visio-5/), the digital counter data was organised into daily averages based on weekdays/weekends, monthly average, busiest day of the week, and busiest days of the period.

### 2.2. Population Survey

This study draws on register data from two population surveys conducted in 2014 (before the path was opened) (*n* = 989) and in 2018 (two years after the opening of the path) (*n* = 2,072) [41,47]. All participants were adult residents in the municipality (≥18 years old). Data were collected using the Select Survey online platform. The population register database is ethically approved by the Norwegian Data Protection Authority (11/00881-2/BSO). The survey includes demographic and socioeconomic data, such as age, gender, income, and education, as well as data on area of residency. It also contains data on self-rated health, life satisfaction, neighbourhood social capital [49], physical activity, use of and satisfaction with availability, and quality of neighbourhood resources (seaside, woodlands, walking/biking paths). The survey asks specifically about the use of the Malvik path, which is the object of investigation in the current study, though the sample is spread across the whole municipality of 168.44 square km. The following variables from the population survey were included in this study: Gender was coded as a binary variable (1 = female and 2 = male). Income (the total household income) was measured categorically based on five intervals, with ‘more than 1.5 million NoK’ as the highest category (one Norwegian krone is approximately10 Euro). Education was measured using five categories ranging from ‘primary school’ to ‘higher education, more than 4 years’. Self-rated health (SRH) was measured as a single item on a four-point Likert scale ranging from ‘very good’ to ‘poor’. SRH represents a validated measure that is widely used (for example, in the large Norwegian HUNT study) (The Nord-Trøndelag Health-study: the HUNT study is one of the largest health studies performed. It is a database of questionnaire data, clinical measurements, and samples from a county’s inhabitants from 1984 onwards. See https://www.ntnu.edu/hunt) and that has been linked to a variety of health outcomes, including mortality [50,51]. Life satisfaction (LS) was measured through a single item, ‘Considering your life at the moment, would you say that you by and large are satisfied with life, or are you mostly dissatisfied?’. Answers were given on a five-point Likert scale ranging from ‘very satisfied’ to ‘very dissatisfied’. This question is a validated measure of life satisfaction frequently used in national and international studies [51]. Neighbourhood social capital (NSC) was measured through agreement on seven statements, such as, ‘I feel safe in my neighbourhood’ and ‘Generally, people do thrive here’ [51]. Answers were given on a five-point Likert scale (‘totally agree’ to ‘totally disagree’) and are indicated by the sum score on all variables (range 7–35, with higher scores indicating higher NSC). Physical activity was measured by frequency (1 = less than once a week, 2 = once a week, 3 = 2–3 times a week, 4 = almost daily), intensity (1 = low intensity, no sweating, 2 = middle intensity, ‘I get short of breath and sweaty’, 3 = high intensity, ‘I exhaust myself)’, and duration (1 = shorter than 30 min, 2 = 30 min to 1 hr, 3 = longer than 1 hr). Satisfaction with the availability and ‘quality of neighbourhood resources’ was measured using single items describing assessments of outdoor facilities (nature contact and seaside facilities), facilities for physical activity, and cycle paths. The respondents were asked to assess the Malvik path specifically. Responses were evaluated on a five-point Likert scale ranging from ‘very satisfied’ to ‘not satisfied at all’. Use of facilities was measured on a similar five-point Likert scale, ranging from ‘not at all’ to ‘very often’. The internal consistency, measured using Cronbach’s alpha, was satisfying for all variables included.

Population data were analysed using SPSS Statistics software (IBM, Oslo, Norway) for Windows, version 22. To examine whether the path was beneficial for health, social inclusion, and physical activity for all groups of citizens, descriptive and comparative analyses were conducted on sociodemographic background variables (including age, gender, household income, level of education, as well as census tract) and on various outcome variables, such as self-rated health (SRH), life satisfaction, thriving in the municipality, neighbourhood social capital (NSC), and physical activity levels (frequency, intensity, and duration). Satisfaction with availability and quality of neighbourhood resources (woodlands, seaside, arena for physical activity, and walking/biking paths) as well as the use of these resources were also described. All these variables were assessed and described both in the survey conducted in 2014 (before the construction of the new path) and in the survey conducted in 2018 (after construction). Comparative analysis was applied to look for changes in these variables between the two times of measurement. Moreover, satisfaction with availability and quality as well as the use of the path were included in the evaluations.

To determine whether all groups benefited from the path and to examine determinants for the use of the path, a descriptive analysis of the group who reported that they used the path ‘very often’ (*n* = 200) was conducted. Additionally, perceptions and use of the path were assessed according to household income to ensure a focus on socioeconomic differences.

Next, included variables were checked for their correlation with the measurement of the use of the path through a Pearson’s correlation analysis. Significant correlations were then included in a stepwise regression analysis. In the first step of the regression analysis, gender, age, education, and income were entered to assess how personal characteristics influenced the use of the path and to apply these as control variables in later steps of the analysis. For the second step, all included variables (age, income, thriving, NSC, satisfaction with availability and quality, as well as the use of nature and seaside facilities, facilities for physical activity, and cycle paths) were entered into the model simultaneously. This was done to highlight and control for relationships between potential predictors and to gain a clearer picture of how the use of the path could be facilitated.

### 2.3. Structured On-Site Interviews

In September 2018, short, structured interviews were conducted along the Malvik path with fourteen adults (male *n* = 3) aged thirty-two to seventy-three (mean age 51.2, SD 14.7). Inspired by a similar study by Schipperijn, Hansen, and Rask [52], the interviewees were asked: (1) How far have you travelled to get to the path? (2) What do you think of the idea of developing the path? (3) What do you particularly like about the path? (4) What do you not like about the path? (5) What do you use the path for? Data from these interviews provided some insight into who the users were (local/non-local residents), their motivation for using the path, and what types of activities they engaged in when using the path. All fourteen interviews were recorded and transcribed verbatim. Transcripts were cleaned, removing unusable ‘fillers’ (i.e., issues that were unrelated to the topic at hand and small words that contain no meaning) [53]. Interviewees’ answers were then summarised and tabulated using summative content analysis [54].

### 2.4. Short On-Site Survey

A short on-site survey was conducted on the path in October 2015 (before the path was officially opened) and in February 2017. Representatives from residents’ associations conducted the survey and approached random users on the path and asked if they would like to participate in the survey. Those who accepted answered the survey on-site and handed it back to the representatives from the residents’ association. In 2015, the survey was conducted during three weekdays, two days during after-work hours (between 16:00 and 19:30), and one during mid-day (from 10:00 to 14:30). In 2017, the survey was conducted on two weekend days (from 11:00 to 14:00) and two weekdays (from 16:00 to 19:30 both days). In the first iteration, there were 142 respondents (54.9% female, mean age 46.4), while in the second iteration, there were 48 respondents (58.3% female, mean age 42.8). This survey assessed demographic data (gender, age, and level of education) and answers to the questions, How often have you used the Malvik path the last seven days? And Why do you use the path? Data were summarised using simple descriptive statistics. The dimensions ‘level of education’ was cross tabulated with how often respondents use the path per week (the fixed options were 1–2 times, 3–4 times, 5–6 times, 7 or more). Respondents’ answers to the open-ended question ‘why I use the path’ were provided in free text where respondents listed several reasons. The reasons for use from the 2015 survey and the 2017 survey were summarised and listed according to the frequency of mention, using a summative content analysis [54]. Similar phrasing was categorised as belonging to the same category. For instance, statements such as ‘because I love the scenery and nature’ and ‘I use it to enjoy the nature along the path’ were put in the same category given the title ‘experience nature and view’.

## 3. Results

The results described in this section draw on the quantitative and qualitative data collected. The results are presented separately according to the type of data collection.

### 3.1. Digital Counter

As shown in Table 1, the number of people visiting the path has increased on both weekdays and weekends since the opening. However, the busiest day (Sunday) and busiest time of the day (13:00) remained identical during the whole data collection period.

### 3.2. Population Survey

Participants in wave I (2014) were on average 47.49 years old and thereby approximately two years younger than participants in wave II (2018) (mean age 49.57). Otherwise, no significant differences were identified. Samples were comparable in respect to gender (61% women in wave I and 59.7% in wave II), large proportions of middle- and high-income groups (with almost 1/3 earning more than 1,000,000 NoK), and very large proportions of participants with higher education, with over half the sample having a university degree and about ¼ having a vocational education (Table 2). Overall, participants in this survey could be described as on average younger, better educated, and with higher incomes than the average Norwegian.

Population survey data indicated that more than three quarters of participants experience good or very good health, while 3.1% (2014) and 2.2% (2018) report poor health. Even higher proportions of participants report high/very high life satisfaction (84.6% in 2014 and 85.5% in 2018) and very good/good thriving in the municipality (88.5% and 86.9%, respectively).

On average, participants were found to have neighbourhood social capital (NSC) levels of 26.26 (SD: 5.18) and 26.65 (SD: 4.94), (range 7–35, with higher scores indicating higher NCS), overall health levels of 3.03 (SD: 0.780) and 3.11 (SD: 0.746), and life satisfaction levels of 4.09 (SD: 0.775) and 4.08 (SD: 0.738) in 2014 and 2018, respectively.

Around three quarters of participants reported that they engaged in physical activity more than once a week and engaged in physical activity intense enough to get sweaty (74.2% in 2014; 73.6% in 2018). More than 9 out of 10 reported that they are physically active longer than 30 min at a time.

No significant changes in respect to health, life satisfaction, thriving, NSC, or physical activity were found in 2014–2018. However, satisfaction with availability and quality and use of neighbourhood resources changed from 2014 to 2018. Among inhabitants, satisfaction with the availability and quality of nature contact and the seaside declined slightly, while satisfaction with availability of sports facilities and walking/biking paths increased. Simultaneously, the use of all these neighbourhood resources increased substantially. Satisfaction and use of the Malvik path itself were first assessed in 2018. Data analysis revealed that 79% were satisfied with availability; 83.4% were satisfied with the quality of the path, and 33.9% reported that they used the path ‘often’ or ‘very often’.

### 3.3. Main Determinants for Using the Path

When describing the fraction of the sample who reported using the path ‘very often’ (*n* = 200) (Table 3), it became apparent that very frequent users were older than the average inhabitant (52.25 vs. 49.57 years old). Almost two-thirds of very frequent users were female. Proximity emerged as a strong predictor for use of the path: 80% of very frequent users lived in the nearest neighbourhood. Frequent users experienced more social capital than the average population, reported that they thrive better, and displayed higher satisfaction with availability as well as the quality of neighbourhood resources. Satisfaction with nature and seaside facilities was higher than satisfaction with sports facilities and walking/biking paths. Especially in respect to the path itself, frequent users expressed high satisfaction: 93.5% of frequent users were satisfied with the availability and 95.9% with the quality of the path (compared to 79% and 83.4%, respectively, in the whole sample). Frequent users of the path also reported use of other neighbourhood facilities more often than average, especially nature facilities (89% reported to use them often) and walking/biking paths (73.6%). They also tended to engage in physical activity more often, but not more intensely or longer than the average participant. No significant differences in health (78.4% good/very good) or life satisfaction (85.6% high/very high) were found between frequent users and other inhabitants.

#### Income

When looking at use and perceptions of the path by level of household income (Table 4), it became apparent that very frequent use of the path decreased with higher household income. A significant drop in proportions of very frequent users appeared among members of households with over one million NoK a year. However, the relationship between income and perceptions of the path are complex. Members of households with the lowest income were the least satisfied with the availability and quality of the path, but they also reported the most use. Members of households with middle incomes seemed to experience the most satisfaction with the path’s availability and quality.

Regression analysis revealed that personal variables (gender, age, income, and education) combined explained 2.4% of all variance in use of the path. Education was not significantly linked to the use of the path. Income emerged as the strongest co-efficient (β = −0.108), which indicated that frequent users of the path were more likely to earn less than average. Frequent users were older than the average user.

In the second step, all variables were entered into the regression simultaneously in order to control for mutual inter-relations. Together, the regression model explained 32.6% of all variance in use of the path. Satisfaction with the availability/accessibility of the path emerged as the strongest predictor for frequent use (β = 0.377), followed by satisfaction with the quality of the path (β = 0.220). Income and use of nature, seaside facilities, and walking/biking paths also emerged as significant predictors (see Table 5), while none of the other included variables became significant at a 0.001 level.

### 3.4. Structured On-Site Interviews

Of the fourteen interviewees, eleven were female. The interviewees’ ages varied from the youngest at 32 to the oldest at 73; the mean age being 51.2 (SD 14.7). Eleven of the fourteen came from the two nearest local areas and had travelled less than two km. The three interviewees who travelled the longest had travelled twelve and thirty kilometres. The interviewees were in general very positive to the idea of developing a path on the old railway tracks. They liked that the path gave access to the seaside and that it was accessible for all people, no matter their physical capabilities. The old path had been narrow, rocky, and uneven and, therefore, probably less accessible for people with physical disabilities.

When asked what they liked about the path, ‘accessibility’ and ‘sea view’ were the most common answers, followed by ‘soft surface’ and ‘flat surface’. Eight interviewees had nothing to say when it came to what they did not like about the path. Among the six who provided answers to this question, two said that the path is too elevated, something that potentially makes it a bit risky for small children (the interviewee was referring to the fact that since the path was laid on top of railway tracks, it is a bit elevated from the ground, some places as much as 40 cm above ground level). Two interviewees referred to a local debate about the name of the path, but without saying that they disliked the name. One interviewee stated that the path lacks lighting, which could be good to have for late-night walks, and one found the path to be too flat and, therefore, not challenging enough.

Most interviewees (*n* = 11) used the path for walking; two mentioned that it was most important for them to meet people there, and one interviewee responded that she used it for exercise, not specifying if this exercise was walking, jogging, or biking, while one reported using the path for jogging.

### 3.5. Short On-Site Survey

In the survey conducted in 2015, 54.9% of the sample were female, while in 2017, 58.3% were female. The average age in 2015 was 46.2 years, 42.8 years in 2017. In 2015, 65.6% had no or low education, while in 2017, only 47.8% had no or low education. Considering the differences between the two samples, a *t*-test shows that gender was significantly different, while age did not differ significantly. The level of education was higher in the later sample, most likely due to the time of day the survey was conducted in 2015 and 2017.

Respondents (*n* = 47 in 2015; *n* = 44 in 2017) provided additional feedback to an optional open-ended question regarding their motivation for using the path. The three most frequently mentioned motivating factors were the same in 2015 and 2017, though the order of the three factors varied. In 2015, the most cited reason for using the path was ‘experience nature and the view’. This was an important motivator in 2017 too, but it was listed second when summarising frequency of mention. In 2017, the main reason for using the path was ‘nice path for walking and good surface’. This was ranked as the number three reason in the 2015 survey. Apparently, what attracted users to the path were contextual features and qualities on and around the path (nature, view, good (soft) surface). Interestingly, ‘exercise and health benefits’, an important reason for use both in 2015 and 2017, dropped from second to third place in importance from the 2015 to the 2017 survey.

## 4. Discussion

The overarching aim of this study was to investigate who uses the path, motivations for use, and what types of activities the path stimulates, as well as to provide knowledge on whether and how accessible green spaces are beneficial for health, social inclusion, and physical activity for all citizens.

A mixed-method research design [48], including a diverse range of relevant and sensitive data and information sources (counting data, questionnaire surveys, registry data, and structured interviews) was applied to capture the complexity of the phenomenon. All in all, this study indicates a significant increase in use of the path from 2015 (before the official opening) to 2018. The path is used by all social groups for various types of activities. Contextual factors such as scenery, sea-view, opportunities to engage in social interaction and surface material are identified as important determinants for using the path. People, in general, are satisfied with the path.

Data from the population survey indicates that the average user of the path is slightly older than the average person in the municipality. Living nearby emerged as a strong predictor for use of the path in both the quantitative and qualitative parts of the study: the population survey indicates that 80% of very frequent users live in the closest included neighbourhood. Perceptions of availability/accessibility emerged as the strongest predictor for use during regression analysis, and they emerged as an important reason for using the path in the analysis of the structured on-site interviews. This correlates with what is found in the literature on determinants for using public green spaces where the distance to green space is the most important factor related to use [19,55,56,57]. Studies have also suggested that the ideal distance is within 300–400 m since longer distance shows rapid decline in use [56]. Studies have also shown that residents’ value greatly and are even willing to pay for having green spaces that are pleasant and easily accessible in their neighbourhoods [58,59,60].

Personal variables, such as income, age, and gender, explained merely 2.4% of all variance in use of the path. This relatively low proportion of explained variance indicates that personal variables (‘who you are’) do not substantially impact the use of the path. The only personal variable that remained significant throughout the last step of the analysis was income, which displayed a negative relationship with use of the path: the less household income participants had, the more likely they were to use the path very often. The literature on use of public green space among lower-income groups is divided; some studies indicate that lower-income groups use public green spaces more frequently because they lack private green spaces in their homes [61], while other studies [15] report that use of green space is lower among lower-income groups because of differential valuation of such spaces for leisure activities. The findings from our study suggest that the path in Malvik is used by a broad range of people in terms of gender, age, education, and other personal variables. This reveals a potential for diminishing the gap in health by offering an arena for physical activity and social inclusion for the less privileged groups of the population.

Entering contextual variables into our analysis raised levels of explained variance substantially (from 2.4 to 32.6%). This is in line with findings from the on-site short survey: while ‘exercise and health benefits’ came high on the list as a reason for use in 2015, this was less important for users in 2017, indicating that the focus had shifted somewhat from personal goals to the appreciation of contextual features. This is supported by the short interview of fourteen persons using the path in 2018, where only one person claimed that she used it for jogging (exercise), while the remaining thirteen said they used it for walking. These results indicate that contextual matters were more important determinants for using the path than personal variables. This is in accordance with previous research underlining the importance of creating supportive social and material settings to promote health and social inclusion [8,40,62,63,64].

Perceived quality of the Malvik path emerged as the second most important predictor of use; matters of quality and aesthetics, as well as opportunities for varied activities, are likely to contribute to increased use of green spaces in urban settings. This harmonises well with previous research, which indicates that vegetation, trees, and the chance to enjoy fresh air and be close to water are factors influencing the use of green space [9,19,20,65]. Such aspects of green spaces might even affect perceptions of safety, as well as whether and how green spaces are used, especially by minority groups [24,62,66]. As perceived quality might indicate the ‘fit’ between features and audience [41], the high satisfaction with the quality of the Malvik path, especially among frequent users, might be understood as a consequence of inclusive processes throughout the planning, design, and implementation of the path and its features. A possible implication of these findings then, is that it might prove beneficial to put effort into the location and design of this kind of outdoor facilities, as this will increase use. Engaging community groups in the planning and implementation of public green spaces has also been recommended as part of stimulating to multi-sectoral collaboration in local communities [67,68]. Thus, our study gains some knowledge into the potential of taking the needs and wishes of residents’ seriously when planning health-promoting interventions.

Even though our study found that people facing socioeconomic disadvantages tend to use the path more than other participants, they are not as satisfied as other groups with the quality of the path: participants living in households with income below 400,000 NoK expressed the least satisfaction with both availability and quality of the path. This might indicate that perceived quality is influenced by whether one has other affordable options available for physical activities [11,12,34,39]. In addition, this could suggest that other factors not included in this analysis are important predictors for use [40], for instance, if one has someone to walk with. Changes in the physical environment that target social determinants of physical activity should, therefore, consider carefully including all groups in the community when planning such interventions, especially those groups that depend more on local resources and offers. This would help local authorities to understand the needs of the residents [68], and research suggests that this can help decrease certain groups’ perceived barriers towards engaging in physical and social activities in the green area [69].

No significant changes in self-rated health, life satisfaction, social capital, thriving, or levels of physical activity on a population level were found between 2014 and 2018. This could indicate that the path did not improve health or health indicators significantly during this period. However, the population survey does not track changes on an individual level, and individual changes might have occurred among the frequent users. Nonetheless, a substantial increase in the use of neighbourhood resources was found. Despite some insecurity linked to this measure (see below), this points towards people indeed being more engaged with their neighbourhood, which might contribute to social connectedness and health benefits in the long run [40]. This is supported by comparative findings [21,49,57,70], which suggest that very frequent users of the Malvik path are more likely to experience more social capital in the neighbourhood compared to the average inhabitant.

Frequent use of other nature and seaside facilities, as well as cycle paths, emerged as important co-efficients of use of the Malvik path. Frequent users also reported engaging in physical activity more often than other participants. This might also point to a well-known dilemma in the domain of public health: people who are already active tend to dedicate new resources to a more active use, thereby widening the gap in health instead of diminishing it [71]. On the other hand, as frequent use of the path might simultaneously affect reported use of cycle paths and nature and seaside facilities (as the path could be described as either of these), these results are difficult to interpret. However, an increase in the use of green spaces in general in the municipality of Malvik indicates that improvement of the path has facilitated more engagement in the neighbourhood.

Comparative analysis indicated that even if frequent users also engaged more often in physical activity, they did so neither more intensely nor for a longer time than the average in the sample. This supports descriptions of the path as a low-threshold amenity that can be used by people regardless of level of fitness or type of activity. This is also supported by findings from the on-site surveys, which suggested that the proportion of people who used the path mainly for exercise had decreased between 2015 and 2018. Hence, the path attracts users of different capabilities with different purposes. The short on-site interviews included utterances such as: ‘I like the benches. I can rest there and, therefore, I can walk the whole path’ (woman, aged 68). A three km pathway can be too long for people less able to walk; hence, a seating area at the halfway point offers the opportunity to sit and relax before resuming the walk. Taken together, these findings support an interpretation suggesting that the path has contributed to more frequent physical activity, as well as more engagement with the neighbourhood, which also might have contributed to increased social capital in frequent users.

## 5. Conclusions

The present paper evaluates how efforts to improve accessibility and quality of a public green space influence use of the green space and how the green space is beneficial for health, social inclusion, and physical activity for residents from various socioeconomic groups in a suburban area. The study builds on a mixed-method design with a broad range of data, both quantitative and qualitative. This implies a variety of challenges and limitations linked to the gathering and interpretations of the data. The population survey, on which part of this study is built, is a cross-sectional survey that does not track individuals but assesses health and health indicators in the municipality on a population level. Thus, changes in outcome variables cannot be linked to individual behaviour over time but must be described at a population level. This makes it impossible to assess how behaviour at one point in time affects health later. However, as health promotion seeks good health for all, being restricted to assessing health on a population and group level might be beneficial: if we cannot explain a lack of impact in some social groups by individual behaviour within the group, the focus is shifted to contextual features and group characteristics. Statistical power could potentially be an issue here; however, the number of survey participants indicates good statistical power, even in subgroups. The lowest number of individuals in a specific subgroup analysed here is *n* = 199, which indicates sufficient statistical power on which to draw conclusions.

The cross-sectional design of the population survey makes it difficult to assess the direction of some relationships. This leads to some uncertainty about how to understand a number of measures. This is illustrated by the above-described difficulties in assessing whether neighbourhood features increase use of the path or whether frequent use of the path influences how one describes one’s use of cycle paths and nature and seaside facilities in general. Next, some specific uncertainty is linked to the measurement of use of neighbourhood resources from 2014 to 2018: while the frequency of use in 2014 was assessed through five specific statements (at least once a month/week/day, etc.), participants in 2018 were asked to describe their frequency of use on a five-point scale ranging from ‘very often’ to ‘not at all’. On the one hand, this might give a less exact picture of the frequency of use and increase uncertainty as to whether levels of ‘very often’ are comparable between individuals and groups. On the other hand, such subjective descriptions might provide a deeper insight into the perceived importance of a feature or activity: if I report using the path very often, this might indicate that it is a very important feature and activity in my life. In any event, the substantial changes in the frequency of use must be interpreted with great care.

Both the short survey and the structured interviews were conducted on-site and provided important information about perceptions, helping to fill some of the emerging concepts with content. Open-ended questions in both the short survey and the interviews allowed respondents to identify in their own words what they conceived to be desirable attributes. On the downside, these interviews include few individuals. Moreover, there was substantial variation as to who was willing to be interviewed on the path: for example, fewer men than women were willing to participate, resulting in a skewed sample. Additionally, as these interviews were on-site, they cannot provide information about who does not use the path. Taken together, even though each method of data gathering and analysis yields specific limitations and insecurities, the broad range and variation of gathered data provide a rich picture of motivation for use, who uses the path, and for which kinds of activities.

Use of the Malvik path cannot be explained by personal characteristics. However, our study found a slight predominance of frequent users belonging to groups facing socioeconomic disadvantages. This finding indicates that the path can contribute to diminishing the gap in health by providing a low-threshold arena for physical activity and social contact. Perceived availability (proximity) to the path emerged as the strongest predictor for its use. However, matters of perceived quality, scenery, sea view, the surface material as well as opportunities to experience nature and engage in social interaction also emerged as important motivators for using the path; hence, contextual factors should be taken into account when evaluating green space effects on health. While these factors are by themselves important for general health and well-being, a secondary long-term effect might be that people are becoming more physically active.

## Figures and Tables

**Table 1 ijerph-16-05015-t001:** Use of the path, measured before (2015) and after (2018) the official opening of the path.

Number of Passings	2015	2018
Monthly Average (*n*)	2226	4448
Daily average (*n*) Weekdays Weekend days	71 53 118	147 126 203
Busiest day of the week	Sunday	Sunday

**Table 2 ijerph-16-05015-t002:** Sample characteristics.

Variables		2014	2018
Gender (% female)		61	59.7
Age Mean (SD)		47.49 ** (13.46)	49.57 ** (15.18)
Income *	<400,000 NoK	11.2	10.3
	400–700,000	27.4	26,6
	700,000–1,000,000	29.9	28.5
	1,000,000–1,500,000	26	27.5
	>1,500,000 NoK	5.5	7.2
Education	Primary school	6.6	7.3
	Vocational education	25.8	25.3
	Secondary school	11.1	11.7
	Higher education, <4 yrs	25.9	24.5
	Higher education, >4 yrs	30.5	31.2

** Significant change 2014–2018 on a 0.001 level, * Income: 1 Euro = approx. 10 NoK. SD = Standard Deviation.

**Table 3 ijerph-16-05015-t003:** Characteristics of ‘very frequent users’ of the path.

Variables		Very Frequent Users (*n* = 200)
Age (Mean, SD)		52.25 ** (14.29)
Gender (% female)		66.0 **
Neighbourhood (%)	Nearest	80.0 **
Thriving (%)		93.0 **
Physical activity	Frequency	91.7 **
	Intensity	73.6
	Duration	96.9
Neighbourhood Social Capital (Mean, SD)		28.11 **(4.94)
Satisfaction with availability	Nature	91.9 **
	Seaside	91.4 **
	Sports facilities	73.4 **
	Walk- and cycle paths	69.6 **
	Malvik path	93.5 **
Satisfaction with quality	Nature	94.9 **
	Seaside	89.7 **
	Sports facilities	70.1 **
	Walk- and cycle paths	65.5 **
	Malvik path	95.9 **
Use of	Nature	89.0 **
	Seaside	63.1 **
	Sports facilities	34.5
	Walk- and cycle paths	73.6 **

** significant difference from the whole sample on a 0.001 level, SD = Standard Deviation.

**Table 4 ijerph-16-05015-t004:** Perceptions and use of the path according to household income.

Household Income	Satisfaction with Availability	Satisfaction with Quality	Frequency of Use
<400,000 (*n* = 199)	73.1	75.2	19.1
400–700,000 (*n* = 515)	83.1	86.7	18.2
700,000–1,000,000 (*n* = 553)	89.3	82.5	16.7
1,000,000–1,500,000 (*n* = 533)	77.2	84.3	7.0
>1,500,000 (*n* = 139)	78.0	81.0	6.8

**Table 5 ijerph-16-05015-t005:** Regression analysis of potential determinants for use of the path.

Step	Entered Variables	Significant Co-Efficient	Variance Explained (%)
1	Gender	−0.072 **	
	Age	0.104 **	
	Income	−0.108 **	
			2.4
2 All variables entered at one step	Income	−0.113 **	
	Use of nature	0.209 **	
	Use of seaside	0.173 **	
	Use of walk/bike-paths	0.115 **	
	Availability Malvik path	0.377 **	
	Quality Malvik path	0.220 **	
			32.6

** = significant at 0.001 level.

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
