# Peer review of "Addressing Inequity: Evaluation of an Intervention to Improve Accessibility and Quality of a Green Space"

_ijerph, 2019, doi:10.3390/ijerph16245015_

Round 1

Reviewer 1 Report

Thank you for the opportunity to review this article. It is important to assess the use of greenspaces, especially by vulnerable groups. This is a very thorough study and, overall, is written and summarized well. I am recommending that the paper be much more concise through some reorganization and removal of some tables. 

Table 1 is reporting results and should be moved to the results section

I am confused by the data analysis section - can this section be placed in either materials and methods or results? 

I think that Tables 6 and 7 can be removed and simply explained in the narrative. 

In general, the text can be much more concise to make the paper shorter and more digestible to the reader. 

Author Response

We thank you for accepting to read our manuscript and for providing insightful and constructive comments that have contributed to improvement of the manuscript.

1) “This is a very thorough study and, overall, is written and summarized well. I am recommending that the paper be much more concise through some reorganization and removal of some tables. Table 1 is reporting results and should be moved to the results section. Table 6 and 7 can be removed and simply described in the narrative”.

Authors’ response:

We have moved table 1 along with the text explaining the content of the table (lines 194-201) to the results section. We removed table 6 and 7, and edited the text related to table 7 slightly due to the removal of this table (lines 415-425). The text now explains how the order of factors motivating the interviewees to use the path changed from 2015 to 2017 (this was only noticeable to the reader when looking at the content of table 7).

2) “I am confused by the data analysis section – could this section be placed in either Materials and methods, or Results?” 

Authors’ response:

We have integrated the text from the Data analysis section with the section entitled Materials and method.

3) “In general, the text can be more concise to make the paper shorter and more digestible to the reader”.

Authors’ response

We have edited the Introduction and tried to make it shorter, more structured and clear. Merging the section on the Data analysis with the Materials and methods and removing superfluous tables have contributed to shortening the manuscript. 

Reviewer 2 Report

This is a well-designed study. The writing is very clear and the theoretical framework is sound. The analysis is straightforward and appropriate. 

Just a couple of minor points for the authors to consider:

1) line 310, it seems like the first cell on the left ("nInter-viviewee") has a typo, it should be Interviewee. 

2) in Table 6, I recommend the authors to use left alignment. It might be easier to read. 

3) Throughout the paper, the authors use "population" but I think the authors meant "sample." 

4) Table 7, have the authors checked the significant difference between sample of 2015 and the sample of 2017 in terms of sociodemographic composition? 

5) Although no significant changes in health were detected between 2014 and 2018, it still might be helpful to show the mean scores of these health measures. 

Author Response

Response to Reviewer 2

We would like to thank the reviewer for taking time to read the manuscript and provide constructive, insightful comments and suggestions. As a result of this feedback, the manuscript has been much improved. 

1) and 2) “This is a well-designed study. The writing is very clear and the theoretical framework is sound. The analysis is straightforward and appropriate. Just a couple of minor points for the authors to consider. 1) Line 310, it seems like the first cell on the left (“nInter-viviewee”) has a typo, it should be Interviewee. 2) In Table 6, I recommend the authors use left alignment. It might be easier to read.

Authors’ response to 1) and 2)

Table 6 has been removed from the manuscript and is now explained in the narrative only (as suggested by Reviewer 1).

3) “Throughout the paper the authors use ‘population’, but I think the authors meant ‘sample’”.

Author’s response to 3): 

We have substituted the term “population” with the term “sample” where this is applicable in the text (in those cases where the term refers to the population survey, and not the general population in the municipality).

4) "Table 7, have the authors checked the significant difference between sample of 2015 and the sample of 2017 in terms of sociodemographic composition?" 

Authors response to 4): 

We checked for significant difference in terms of sociodemographic composition and included what we found in the manuscript (lines 411-413). The following text has been added:

“Considering the differences between the two samples, a t-test shows that gender was significantly different, while age did not differ significantly. The level of education was higher in the later sample, most likely due to the time of day the survey was conducted in 2015 and 2017”.

5) “Although no significant changes in health were detected between 2014 and 2018, it still might be helpful to show the mean scores of these health measures”.

Authors response to 5):

We checked for significant changes in the overall health, life satisfaction and social capital and added this information in the manuscript, lines 322-325 (the edited text highlighted in yellow):

(lines 322-325): “On average, participants were found to have neighbourhood social capital (NSC) levels of 26.26 (SD: 5.18) and 26.65 (SD: 4.94), (range 7–35, with higher scores indicating higher NCS), overall health levels of 3.03 (SD: .780) and 3.11 (SD: 7.46) and life satisfaction levels of 4.09 (SD: .775) and 4.08 (SD: .738) in 2014 and 2018 respectively”.

Reviewer 3 Report

It is a very interesting paper. The methods are clear and results indeed provide some new insights into the issue. However, I find that the literature review in the introduction part does not involve the latest papers in recent years. I thus recommend update this part to make the paper more attractive. I also provide some references for your consideration. Finally, I am thinking the limitation part can be integrated with the conclusion part or just delete the conclusion pat. In most cases, the discussion can be integrated with the conclusion part.

(1)  Impact of community deprivation on urban park access over time:
Understanding the relative role of contributors for urban planning.  Habitat International 92 (2019) 102031
(2)  Public health in linkage to land use: Theoretical framework, empirical
evidence, and critical implications for reconnecting health promotion
to land use policy.  Land Use Policy 57 (2016) 605–618
(3)  Social inequalities of park accessibility in Shenzhen, China: The role of park quality, transport modes, and hierarchical socioeconomic characteristics.  Journal of Transport Geography 62 (2017) 38–50.

Author Response

Response to Reviewer 3 

We would like to thank you for accepting to read our manuscript and for providing insightful and constructive comments that have contributed to improvement of the manuscript. We also appreciate the literature recommendations provided.

1) “It is a very interesting paper. The methods are clear and results indeed provide some new insights into the issue. However, I find that the literature review in the introduction part does not involve the latest papers in recent years. I thus recommend update this part to make the paper more attractive. I also provide some references for your consideration”.

Authors response to 1):

We have revised the Introduction, improved the structure and tried to make it more concise and clearer. We have also added some more up-to-date literature. Literature added:

Engemann, K.; Pedersen, C.B.; Arge, L.; Tsirogiannis, C.; Mortensen, P.B.; Svenning, J-C. Residential green space in childhood is associated with lower risk of psychiatric disorders from adolescence into adulthood. PNAS. 2019, 116, 5188-5193.

Sugiyama, T.; Carver, A.; Koohsari, M.J.; Veitch, J. Advantages of public green spaces in enhancing population health. Lands. Urban Plan 2018, 178, 12-17.

Xu, M.; Xin, J.; Su, S.; Weng, M.; Cai, Z. Social inequalities of park accessibility in Shenzhen, China: The role of park quality, transport modes, and hierarchical socioeconomic characteristics. Journal of Transport Geography2017, 62, 38-50.

2) “I am thinking the limitation part can be integrated with the conclusion part or just delete the conclusion pat. In most cases, the discussion can be integrated with the conclusion part”.

Authors response to 2):

We have included the ‘Methodological limitations’ in the ‘Conclusion’.

Reviewer 4 Report

This study explores the benefits of greenways and the factors that may influence greenway use, which can provide some empirical evidence. However, I suggest the authors to further think about the theoretical contributions and the novelty of the study. What is the relationship between conclusions from this study and findings of existing study. Have the issues addressed in this study been somehow explored before?

line.104 You mentioned the renewed pathway was opened on June 2016 and the digital counter was installed on October 2015, and the pathway was opened on June 2016. I was wondering when the renew project was finished. Was the greenway under construction when the digital counter was installed? What is the model of the digital counter. You may need to elaborate more on how the counter was working.

line109. Where do the samples live in. Are they within the service radius of the green way?

Line 139. Is the greenway visitation specifically mentioned in the survey? Or participants just report their overall natural environment visitation.

Line 155. Does the interview only involve 14 people?

Line 223 Where was the counter installed? How many accessible points the greenway have?

Author Response

Response to Reviewer 4

We would like to thank the reviewer for taking time to read the manuscript and provide constructive, insightful comments and suggestions. As a result of this feedback, the manuscript has been much improved. 

1) “This study explores the benefits of greenways and the factors that may influence greenway use, which can provide some empirical evidence. However, I suggest the authors to further think about the theoretical contributions and the novelty of the study. What is the relationship between conclusions from this study and findings of existing study. Have the issues addressed in this study been somehow explored before?”

Authors response to 1):

We have tried to cater to these shortcomings without making the manuscript too much longer. We have tidied up the introduction and referred to studies investigating issues that we are addressing and investigating in our study. We have also added a short section at the end of the conclusion where we highlight more clearly what the novelty of our study is. 

(Lines 628-632) “Lastly, our study should provide some new insights into the significance of suburban public green spaces. Much research has been devoted to investigating the health and environmental benefits of creating greener cities. Our findings indicate that easily accessible and pleasant, high quality green areas have the potential to improve the health and wellbeing of residents in suburban areas as well”.

We have also added references to other studies reporting on similar findings (added text highlighted in yellow):  

(lines 609-620) “Our study indicates that it might prove beneficial to establish green spaces near living areas and that putting effort into the location and design of outdoor facilities is important. This has been supported by other studies that have shown that residents’ value greatly, and are even willing to pay for, having green spaces that are pleasant and easily accessible in their neighbourhoods [51, 52, 53]. This, in turn, can be taken as an argument for inclusive processes, ensuring that a local green space displays the qualities desired by the local community, especially by inhabitants belonging to disadvantaged groups. The active involvement of community groups in creating supportive physical and social environments have been found to facilitate greater understanding of residents’ needs [52] and has been recommended as part of stimulating to multi-sectoral collaboration in local communities [54, 55]. Thus, our study gains some knowledge into the potential of taking the needs and wishes of residents’ seriously when planning health promoting interventions”.

2) “Line.104 You mentioned the renewed pathway was opened on June 2016 and the digital counter was installed on October 2015, and the pathway was opened on June 2016. I was wondering when the renew project was finished”.

Authors response to 2):

We have added some more information regarding the opening of the path, in lines 95-96 in the manuscript (the added text is highlighted in yellow):

“The new, improved path was officially opened in June 2016, though it was opened for use approximately two months prior to this”.

3) “Was the greenway under construction when the digital counter was installed? What is the model of the digital counter? You may need to elaborate more on how the counter was working”.

Authors response to 3):

We have added some more information regarding the digital counter and specified when it was placed along the path and that it was placed there before the path was completed and officially opened. The edited/added text is highlighted in yellow (from lines 114-124 in the manuscript):

“A digital counter, PYRO box from Eco Counter [https://www.eco-compteur.com/en/resources/], that registers each passing at the path was set up in October 2015, prior to the official opening of the renewed path (June 2016). The PYRO-Box is a people counter using passive-infrared, pyroelectric technology that is suitable to count both cyclists and pedestrians. The digital counter has registered the number of passings on the path each day since October 2015, and all registrations from then through to the 31st of December 2018 were included in the study (altogether 37.5 months). By use of Eco-Visio data analysis platform 5 [https://www.eco-compteur.com/en/produits/eco-visio-range/eco-visio-5/], the digital counter data was organised into daily averages based on weekdays/weekends, monthly average, busiest day of the week, and busiest days of the period”.

4) “Line109. Where do the samples live in. Are they within the service radius of the green way?”

Authors response to 4):

We added information on this in lines 139-140:

“…the sample is spread across the whole municipality of 168,44 square kilometres.”

5) “Line 139. Is the greenway visitation specifically mentioned in the survey? Or participants just report their overall natural environment visitation?”

Authors response to 5):

We added information on this in lines 137-139:

“The survey asks specifically about use of the Malvik, including the path that which is the object of investigation in the current study).”

6) “Line 155. Does the interview only involve 14 people?”

Authors response to 6):

We think that this is specified in the current manuscript:

Lines 205-206: “In September 2018, short, structured interviews were conducted along the Malvik path with fourteen adults (male N=3) aged thirty-two to seventy-three”

Lines 385-387: “Of the fourteen interviewees, eleven were female. The interviewees’ ages varied from the youngest at 32 to the oldest at 73, the mean age being 51.2 (SD 14.7). Eleven of the fourteen came from the two nearest local areas and had travelled less than two kilometres.”

Round 2

Reviewer 1 Report

Thank you for your revisions and addressing my specific comments. 

Author Response

Response to reviewer one: 

Thank you for taking time to read our revised manuscript. 

Reviewer 4 Report

Please review literature in green spaces' benefits in suburban areas and re-think the contributions of the study. The author may need to elaborate on the possible reasons that suburban residents may receive different kinds of benefits from green space  than urban residents? If no evidences support such a difference, the authors may need to think about the contributions and justifications of the study.

Author Response

Response to reviewer four: Please see the attachment. 
